# Navigating Religion Online: Jewish and Muslim Responses to Social Media

**Jauhara Ferguson \*, Elaine Howard Ecklund and Connor Rothschild**

Department of Sociology, Rice University, Houston, TX 77005, USA; ehe@rice.edu (E.H.E.); clr6@rice.edu (C.R.)
\* Correspondence: jf56@rice.edu

**Abstract:** Although social media use among religious communities is proliferating, significant gaps remain in our understanding of how religious minorities perceive social media in relation to their faith and community. Thus, we ask how individuals use religion to frame moral attitudes around social media for Jews and Muslims. Specifically, how does social media shape understandings of community? We analyze 52 interviews with Jews and Muslims sampled from Houston and Chicago. We find that Jews and Muslims view social media as a "double-edged sword"—providing opportunities to expand intracommunal ties and access to religious resources, while also diluting the quality of ties and increasing exposure to religious distractions. These findings help us understand what it is about being a religious minority in the US that might shape how individuals engage with social media. Moreover, they suggest that social media may be transforming faith communities in less embodied ways, a topic that is of particular relevance in our pandemic times.

**Keywords:** Jew; Muslim; morality; religion; social media

## 1. Introduction

The advent of social media technologies has made it more possible than ever for people to connect across different regions, demographics, and worldviews, in times of crisis and normalcy. The widespread impact of the global COVID-19 pandemic, in particular, has compelled many individuals and organizations to increase their online presence, forming alternative, less embodied ways of gathering. In the United States, despite the increased controversy over privacy concerns on social media platforms such as Facebook, media use remains constant (Perrin and Anderson 2019). Although some research has explored the interplay between religion and social media use generally (Campbell 2012b; Cheong 2017; Hjarvard 2016; McClure 2017), less research has examined patterns across religious traditions, especially traditions that have a significant presence within the United States but are still a numeric minority.

About 20 percent of US adults share their faith identity online, and nearly half of US adults see religion shared regularly by others online (Pew Research Center 2014). Significant gaps remain, however, in our understanding of the interplay between religion and social media. Specifically, we have little understanding of how religion may shape individuals' moral attitudes about use of digital technology and gathering online or how social media may be transforming a sense of religious community for those religious minority groups who are often marginalized in society. For some religious communities, social media may present alternative ways to build or maintain community in the face of widespread societal marginalization. For religious minorities such as Jews and Muslims, in particular, social media may also be used as a safe haven distanced from the societal pressures of anti-Semitism and Islamophobia (Marizan 2016). Social media may also aggravate marginalization, as chat rooms and other platforms provide space for greater interreligious conflict and polarization (Alvstad 2010; Neumaier 2020). Yet, limited work interrogates the ways in which members of these minority religious traditions use digital media technologies and how they perceive social media in relation to their faith.



Here, we address this gap by asking the following questions: To what extent do Jews and Muslims use religion to frame moral attitudes around social media? Moreover, how does social media shape understandings of community? We draw on data from a broader study, focusing particularly on interviews with 52 religious minorities (29 Jews and 23 Muslims), comprising both leaders and congregants, and sampled from religious organizations in Houston and Chicago. We find that Jews and Muslims perceive social media as neither exclusively providing social benefits nor exclusively evoking moral concerns. Instead, community perceptions of social media are complex. While social media allows for connectedness and greater access to religious resources, our respondents explain that it also presents new distractions and temptations that may test one's ability to stay on a moral path.

### 1.1. Religion and Social Media

In recent years, public discourse around the ways in which digital technology and social media is changing religion has amplified. Scholars have examined mediatization, or the long-term processes between media and social and cultural change (Hjarvard 2016; Lövheim 2011; Lövheim and Lynch 2011; Mishol-Shauli and Golan 2019; Rota and Kruger 2019). Mediatization not only examines media in relationship to long-term social and cultural processes, but it also provides theoretical explanations for the resurgence of religiously based social and cultural change through new digital media and the increased visibility of religion across many public institutions (Hjarvard 2016; Rota and Kruger 2019). As public religious figures, such as Pope Francis, join popular social media platforms such as Instagram (Stack 2016), there are greater opportunities for scholars to examine the effects social media has on both encouraging the adaptation of religion and challenging religious communities within an increasingly global society.

McClure (2017, p. 481) draws on the idea of "tinkering" to describe the way digital media changes how people relate to religion. Previous studies used the concept of tinkering to refer to the way younger generations think about religion and how the advent of internet technologies causes people to recreate the way they view themselves (Berger et al. 1974; Turkle 1997; Wuthnow 2010). McClure (2017), however, extends this concept to examine the ways in which internet usage can potentially shift religious identities. In other words, the internet can be an apparatus through which religious individuals and groups may "tinker" with long-held conceptions of religion, providing the digital space for religious individuals to be more religiously inclusive (McClure 2017). The increase in social media use among religious people of diverse religious, racial, and ethnic backgrounds may foster increased interaction across religious and racial-ethnic boundaries and, in turn, decrease the likelihood of religious exclusivity (McClure 2017).

There is also a push within scholarship on digital media and religion to distinguish between religion online and online religion. Frost and Youngblood (2014), for example, state that religion online describes the online functions of traditional religious communities. This includes the way that religious communities use digital media platforms to reiterate religious community, faith identity, and community outreach on behalf of communities that primarily exist within physical religious spaces (Campbell 2012c; Frost and Youngblood 2014). Online religion, however, refers to religious and spiritual practice that is done using the conduit of the internet (Frost and Youngblood 2014). These distinctions in the relationship between religion and digital media point to how online community spaces can serve as both a supplement and substitute to physical religious communities (Campbell 2012c). It is important to understand the relationship religious individuals and communities have to digital media because it may provide more insight about the benefits and challenges of maintaining an embodied religious community.

### 1.2. The Virtual Religious Community

According to Durkheim, the collectiveness of religious practice is what drives individual belief. The expression of religion on social media may shift the boundaries of

the "collective", as internet technologies engage social networks in a way that usurps physical community and geographic boundaries (DiMaggio et al. 2001). Recent work on megachurches and virtual religious communities has already begun to complicate the notion of physical presence as a requirement to achieve a sense of community (Campbell 2012a; Campbell and Vitullo 2016; Hutchings 2017). Juergensmeyer et al. (2015, p. 43) argue that virtual mediums have begun to assist, if not replace, traditional "brick and mortar" religious institutions. The challenges of modernity and globalization have created new opportunities to expand the way religion is conceived and practiced within society (Juergensmeyer et al. 2015). This can be particularly true for some religious minorities because of the way the internet allows for underrepresented groups to create meaning from their marginalized societal position in the process of reinforcing religious identities. Warren (2018) describes the way that British Muslim women, in particular, use digital media to create content for other Muslim women through lifestyle media that specifically fosters a sense of "Muslim woman" identity formation. For these women, religious marginalization within British society leads them to respond with specifically curated platforms that emphasize their identity and sense of belonging. In this way Muslim women use social and digital media to forge new communities cross-nationally. Social media then can expand the definition of the religious collective and shift the way individuals within this collective are connected to each other, as well as how they relate to one another when they are meeting face to face. These examples also demonstrate the intersections between digital media, religion, and gender. For example, some scholars have written about how social media allows for greater communication internationally and therefore can redefine what it means to "do" feminism (Baer 2016; Salime 2014; Tsuria 2020; Zebracki and Luger 2018). This not only has implications for the shared experiences of individuals within a religious community but could also potentially undergird moral concerns about technologies that have facilitated these changes.

While social media can expand connections and create religious identities within religious communities, social media can also expand interreligious contact between religious communities (Neumaier 2020). Social media can potentially connect individuals of diverse religious communities and increase interreligious dialogue. Scholars argue, however, about whether this potential for contact actually increases productive interreligious engagement or if it increases conflict between religious groups (Alvstad 2010; Neumaier 2020; Tsuria 2013). Despite the potential for internet technologies and social media to expand conceptions of community, some scholars caution against the conclusion that this potential for greater connectedness results in a more engaged public. Although social media provides individual users the opportunity to come together as a group in a "public" space, Kruse et al. (2018) argue that Millennial and Generation X users do not use social media in a way that engages the "public sphere". The public sphere is defined as the place where individuals meet openly to exchange ideas and ultimately fuel political change (Habermas 1991; Kruse et al. 2018). These generations of social media users avoid political and religious discourse on internet platforms (Kruse et al. 2018). This is due to factors such as increased online privacy concerns, fear of consequences from employers, increased engagement with others of similar political backgrounds, and the perception that social media is a space for positive interaction (Jenkins 2006; Kruse et al. 2018).

These findings also allude to the idea that the internet as a public space restricts information access almost as much as it shares it. Internet algorithms on social media sites often promote certain clusters of social media users to see select sets of information based on their browser history (Jenkins 2006; Kruse et al. 2018; Loader and Mercea 2011; Trottier 2011). This suggests that the potential for a more expanded and engaged virtual religious community could be limited due to the constraints of being online. While religious communities may be able to virtually connect with individuals who are regularly involved in physical communities, they may struggle to connect with newer members or those who do not want to disclose their religion online. Additionally, individuals who lack the technological resources to connect with virtual religious platforms will be even

further isolated. Social media has the potential to both promote and stifle engagement within religious communities. While social media can expand the broad reach of religious communities, the extent of its reach may depend on technology itself.

### 1.3. Jews, Muslims, and Morality Online

While the internet does not guarantee the expansion of the public sphere, the internet and social media provide the potential to build diverse social networks. This might create, however, ramifications for religious ethics and moral concerns. For religious communities, the internet's potential to steer believers away from the straight and narrow presents a challenge in fully embracing the platform. Although scholars have recently started looking at the effects of the internet on conceptions of community, religiosity, and adolescent spirituality (Bobkowski and Pearce 2011; Campbell 2012b; Juergensmeyer et al. 2015; Kruse et al. 2018), little research has specifically examined the effects of social media and internet technologies on religious morality, and even less research has explored such effects on minority religious communities.

The minority religious communities we focus on—Jews and Muslims—may approach social media in different ways; namely how they engage with social media, how it complements their religion, or their moral attitudes toward it. Studies regarding Islamophobia on Facebook have found a general trend toward more hateful social media content over time (Auxier 2021; Awan 2016; Törnberg and Törnberg 2016). Similarly, Finkelstein et al. (2018) have documented the rise of anti-Semitism online, especially following high-profile political events. Although some research has explored the role of social media platforms as safe spaces for sexually marginalized youth (Lucero 2017), no such research documents online media as a similar platform for religious communities in the face of marginalization.

Hitlin and Vaisey (2013) argue that sociology should begin to engage more in the study of morality, as religious communities each have their own moralities. Religious communities online create conceptions of morality aligned with their religious beliefs. For religious minorities, this challenge can be particularly salient. In her study of Muslim women's political activism in Indonesia, for example, Rinaldo (2013) argues that Muslim women approach the challenge of pornographic online content from their respective approaches to understanding Islam, as well as their differing historical and sociopolitical contexts. Muslim and Jewish moralities are not monolithic but are dependent on environment and religious frameworks.

In a more recent study, Al-Rawi (2015) outlines the online responses of Arabic speakers on YouTube to controversial Danish cartoons of the Prophet Muhammed. Although Al-Rawi (2015) does not directly address the concept of religious morality online, his analysis of the responses demonstrates the distinct challenges that religious minority communities face as a marginalized group online. The Islamic religious moral injunction against depicting prophets presented a unique challenge for Muslim communities online and offline (Al-Rawi 2015). Muslim internet users utilized specific religious frameworks to virtually retaliate against what they considered to be immoral, while offline Muslim communities struggled to combat extreme interpretations and responses to this moral injunction by online and offline Muslims (Al-Rawi 2015). The presence of social media may provide specific challenges to religious individuals within individual communities, especially as they interact with competing moralities online.

### 1.4. Research Questions

The present study contributes to two core empirical gaps: first, it fills an enduring gap in the extant religion-science literature on religious minorities (see Vaidyanathan et al. 2016); and second, it addresses the gap in our understanding of the interplay between religion and social media in shaping understandings of community more broadly. This study asks two questions to fill current gaps in the religion-science literature. First, to understand the complicated relationship religious people may have with social media

technologies, we ask the following: How does religion frame moral attitudes around social media for Jews and Muslims? Second, considering the pivotal and unique role the internet must play in connecting religious people: How does social media shape understandings of community? Coupled with more recent online data regarding how these communities responded to COVID-19, we find that social media is laden with both opportunities and risks. Jewish and Muslim communities use social media to build greater connectedness and bridge generational gaps, while also acknowledging potential moral distractions and temptations. Nonetheless, in the current era of social distancing, these communities find social media technologies essential for maintaining community.

## 2. Results

After examining the interviews with Muslim and Jewish respondents, we found a general overlap in our results. Responses tended to fit within one of three categories: perceived social benefits and drawbacks of social media, moral concerns, and shifting the boundaries of communities. Often, a respondent would touch upon all three throughout the interview, capturing the complicated nature of social media and religion. These themes surfaced for both Muslim and Jewish respondents, although at times they were expressed in ways unique to the dynamics of the particular community. Respondents saw potential benefits of social media through the way it could expand one's knowledge of religion. Both communities expressed concerns about social media's impact on the quality of community relationships and the influence on religious morality. Lastly, both Jewish and Muslim respondents believed that social media could expand the conception of community by increasing access to those who may not visit physical religious spaces.

### 2.1. Perceived Benefits and Drawbacks of Social Media

Respondents expressed that social media could lead to spiritual growth, if used in the right manner. This was the case both personally and organizationally. For individuals, technology and online media could be used to learn more about religion, as was the case with this Muslim respondent[1]:

> Actually, technology is helping [me] understand my faith better ... As long as you are using technology for humane purposes, it's great. If you try to use technology to create harm to others, then there's a conflict.

For this respondent, social media is a double-edged sword. So long as it is used for "humane purposes," such as education, technology and media can be helpful tools for growing deeper in one's faith. This respondent asserts that digital media can aid his understanding of his faith because of the ability he has to search religious texts online, listen to religious lectures, and connect with people within his faith community through different social media platforms. Outside of the individual religious experience, social media can also inform *others*. An illustration of this comes from another Muslim respondent[2] who noted that online media could be a tool to inform others about his religion:

> ... [Y]ou can always go on YouTube and find ... videos from Islamic speakers, kind of like invite people to Islam, talk a little bit about—even the scientific stuff you'll find on YouTube.

This sentiment was echoed by numerous respondents, including both Jews and Muslims. One Jewish respondent[3] said that social media led people to be "connected in all kind of places ... getting their Judaism ... their insights, their questions answered, everywhere and anywhere."

For these respondents, social media is viewed as a multifaceted tool, which allows for personal edification, proselytization, and advertisement. Social media allowed those

---

[1]  Int4, Muslim, Male, 50+.

[2]  Int3, Muslim, Male, 20.

[3]  Int12_RL, Reform Jew, Male, 50.

interested in faith to learn more about religion through both resources available online and through social networks.

As often as respondents praised social media for creating community, they also denounced it for diluting the quality of relationships. Both Muslim and Jewish respondents acknowledged that social media has completely changed interpersonal relationships and communication. People within religious communities may opt to communicate information via social media platforms, which may alienate some members. Not everyone within a single "community" may have a social media presence online. Additionally, respondents were critical of the way that community members may not always personally interact with each other and instead rely on virtual communication to build community. One Orthodox Jew in his 30s[4] said that social media both accelerates the rate of and lowers the quality of communication:

> ... the irony of the lack of connectivity that we have, because of all of our lack of human interactions and relying on social media and texting, you know, I think has lowered the quality of relationships and general happiness in the world, in my opinion.

This respondent does not disagree with the premise that social media allows for *more* communication; rather, he is arguing that that communication is simply worse than it once was. Social media, in his view, has replaced face-to-face interaction. His response signifies broader anxieties among some respondents who identified social media communication as an overall societal trend that devalued the significance of face-to-face interaction. A Muslim respondent[5] echoed that same argument when he said the following:

> ... the time which people need to spend with each other, they spend on technology ... it connects people, but it really does not create solid relationships. Because solid relationships require human interaction and human conversation and human dealings and that decreases and only technological connections, they increase.

Although these responses are not specifically related to religion, the respondents are voicing disappointment in social media for its effects on human relationships more generally. This conveys tensions surrounding the importance physical interaction should and does play within religious communities.

The benefits and drawbacks of social media are closely related. Social media creates the possibility for broadening the scope of community to become more inclusive to those demographics who may not have had the same access to physical religious spaces. The elderly, disabled, and frequent travelers may all gain some benefit from the adoption of a virtual community. In comparison, millennials may generally connect more easily and gain greater access to religious materials and resources online. Muslim and Jewish respondents alike saw social media platforms as a way of directly connecting congregants to online resources that could answer questions about their faith. In this way, Jewish and Muslim respondents "tinkered" with the idea of religious authority. Whereas traditionally faith leaders may be directly approached to answer questions, comment on polemical issues, or advise congregants, Muslim and Jewish congregants could seek answers via social media platforms such as YouTube or Facebook. For our respondents, this was mostly in addition to the leadership given within their communities. On a broader level, however, religious authorities online could have the potential to replace in-person religious authority in lieu of access to physical religious resources. Social media also increases the amount of choice one has in the types of religious leadership and perspectives they seek. Depending on the tradition, this has the potential to reform established mechanisms of obtaining religious knowledge and to reinterpret the meaning of religious authority.

---

4    Int2, Orthodox Jew, Male, 36.
5    Int1, Muslim, Male, 58.

### 2.2. Moral Concerns about Social Media

Other respondents talked about moral concerns with social media. More specifically, social media was said to expose individuals to morally objectionable content and distract them from religious experiences and expressions. For two Orthodox Jewish respondents, social media challenged conceptions of modesty. One Orthodox Jew[6] argued that social media exposed children to "immodesty" and "garbage":

> ... [Children] don't need to surf the Internet ... The Internet ... uses tons of immodesty, and you never know what's going to pop up, and there's violence and things that they don't need to be exposed to. They shouldn't be.

For this respondent, social media may pose problems when it exposes individuals to objectionable content. In a similar vein, another Orthodox Jew[7] said that social media makes people become egocentric and less modest:

> You know, we teach modesty. So, people normally think of modesty in terms of ... dress, but modesty is also like, don't talk about yourself so much, and don't—there are things you share with people. But people get on Facebook now and they have two-hundred followers. ... I mean, there's nothing wrong with that, but how egocentric could you be? Do you really believe that two-hundred people want to hear about it? But this is our society.

This respondent alludes to the broader issue of morality online. Because social media has the potential to expand one's social reach, it can also contradict the emphasis placed on humility or modesty within the Orthodox Jewish tradition. Social media becomes a virtual space in which everyone can call attention to themselves. This not only raises concerns about religious morality but may also complicate social relationships and issues of mental health. We of course are not arguing that these distinctions are based on the faith tradition per se, but from this excerpt and others in the interview, the respondent seems to be drawing on pieces of his faith tradition (in addition to other potential factors) to explain the morality of social media use.

Three Muslim respondents explained that social media can be a distraction in direct competition with religious practices. One Muslim respondent[8] provided an example of social media—or technology more generally—disrupting otherwise meditative religious experiences.

> In some ways, I would say ... [smart phones] limit our religious experience ... because they're a constant source of distraction. In the mosque, always someone's phone is ringing despite, you know, many announcements. So, and if it rings or if it's in vibration mode, then still you are focused 'who is calling' you know? So, they do bring distraction, in terms of ... devotion which religion requires in worships and meditations.

Another Muslim respondent[9] echoed this sentiment while challenging the notion that social media can bring religious people closer to their religion:

> I think social media can distract from your religious practice ... some people say that social media is great because it keeps me in touch with all this religious stuff and all these religious people that I would normally not have access to. And that can be great, but I think it's cheap. I think real religious enlightenment takes effort. It takes work. You have to engage in the [religious] texts. You have to challenge yourself mentally to do it. You can't just—I think social media can often reduce stuff like learning.

---

6   Int10, Orthodox Jew, Female, 39.
7   Int1, Orthodox Jew, Male, 58.
8   Int1, Muslim, Male, 58.
9   Int17, Muslim, Male, 35.

These responses point to the way media technologies might impede religious worship for Muslims. Although social media itself may not restrict prayer, receiving multiple notifications from a social media app may quite literally distract one's focus from ritual prayer. The acknowledgement that multiple announcements are given to silence phones before communal prayer signifies the way that social media has the potential to change the dynamics of religious practice. Religious rituals that were traditionally practiced in particular ways may slowly shift to accommodate these changing dynamics.

Finally, a few respondents expressed more unique concerns about social media. One respondent, a Muslim male in his 50s,[10] claimed that technology "can be misused and it can create the problem of authenticity of information." Another, this time a young Muslim woman,[11] expressed her fear of social media as a platform for bullying and hate, claiming that social media technologies do not foster dialogue but, instead, bigotry and one-sided rants.

These respondents represent a spectrum of *level* of concern, and a wide variety of *type* of concern, related to social media technologies. Although some concerns—such as the fear that social media would expose children to objectionable content or distract from religious experiences—were explicitly religious in nature, others—such as the fear that social media can spread inauthentic information—were far from religious concerns.

### 2.3. Changing Dynamics and Boundaries of Communities

According to many of our respondents, social media had the power to shift community boundaries. By eliminating geographic barriers, social media provided a pathway toward transnational community building and connectedness. These global connections are important for religious minorities, who may be distanced geographically from their religious counterparts. As one Reform Jewish respondent[12] put it:

> [O]ne of the things that's great about social media is that it breaks down a lot of traditional boundaries, like geography, for example. People are able to connect with other people who live thousands of miles away in a way that they didn't used to before . . .

For this respondent, social media allows connection with religious peers across the globe. In that way, social media has the power to form new relationships and build global religious ties.

Social media has also changed the dynamics *within* existing religious communities. One Rabbi[13] made note of his recent decision to put his congregation's worship services on the internet:

> [W]e now stream all of our services, our worship services, which in some ways is great because we can reach more people. There are people who are in the hospital who have a chance to feel like they're part of it, or in the military, or at home incapacitated—lots of things. Members who go away to college or move out of town, I mean there's lots of ways that people can access it that they couldn't before.

Such changes may come at a cost. Although online services, and social media more generally, allow for *more* connection, this connection may be less *meaningful* than it once was. For the Rabbi at hand, the "obvious flipside" is that "there is no sense of 'this is our community.'" Because physical boundaries have been nearly eliminated, he asks, "what does it mean to be 'us?'" Another Rabbi[14] agreed:

> In an age where we're always connected, studies show that we're more disconnected now than we've ever been. It's kind of a double-edged sword. We've talked a lot about "What does that mean?" Do we have an online community?

---

10 Int1, Muslim, Male, 58.
11 Int6, Muslim, Female, 22.
12 Int18, Reform Jew, Female, 39.
13 Int10_RL, Reform Jew, Male, 30.
14 Int11_RL, Reform Jew, Female, 36.

... [H]ow do we approach this new online community? What does that mean for our congregation?

An apparent tension arises between *embodied* and *virtual* community. Although a virtual community allows for a greater number of community members, some religious leaders believe that digital communication hinders the *quality* of community and of relationships. As religious rituals have historically occupied physical spaces, the gradual transition toward digital mediums may pose a unique challenge for the future of religion. In response to this tension, some religious communities construct new boundaries that make the importance of being a committed "member" more important to congregants. An example of this would be a Reform Jewish community[15] setting limits and password protections for certain online content, which helps answer the question, "What does it mean to be 'us?'"

Another issue with respect to shifts in community that arose was regarding generational divides between "digital natives" and "digital immigrants". Several respondents—mostly older respondents—expressed anxiety that social media was a "young" phenomenon that they did not understand. Others questioned whether their congregation would be able to adapt to and "survive" social media. A final group of respondents expressed their willingness to adapt to social media in order to cater to younger populations. Of all those sentiments, the most candid were ones of confusion. One Muslim man in his late 50s[16] captured this quite succinctly:

> Right now ... all the young people are very proficient with social media. The older generation have no clue what social media is and that is creating a big communication gap between the younger people and the older people.

Another man in his late 50s, this time a Reform Jew[17], said the following:

> You know, I'm an old guy ... I don't know how all that stuff—I mean I know how it works; I understand the concepts—I don't use it ... I understand on an intellectual level that ... this is the preferred communication platform of today. Will I be adept at it? No.

Responses such as these two often differed in their confidence to "understand" social media technologies. While some religious people view themselves as capable of adoption but unwilling, others were shut out of social media use because of lack of proficiency, creating what some congregants called a "big communication gap" between generations, with few opportunities to bridge that gap. Others painted social media technologies as just another generational shift. One Rabbi[18] told us that his Reform Jewish congregation was asking "Where are the millennials? Where is everybody going?" to which he responded as follows:

> I'm as worried as anybody else, but this isn't the first time we've had institutional shifts or sociological changes ... we will have to listen and engage technology and social media and the debates of our times in a living thriving Reform Judaism.

Social media, then, creates a generational divide that evokes different responses from different religious people. The questions of *whether* to engage, and *if it is possible* to engage with a younger generation and bridge the digital divide, endure in the minds of many of our respondents.

### 3. Discussion

Our findings suggest social media provides neither benefits nor concerns alone but is instead individually negotiated by both Jewish and Muslim respondents in ways consonant with their traditions, with their understandings of community, and with their moral framing

---

15    Int11_RL, Reform Jew, Female, 36.
16    Int4, Muslim, Male, 50+.
17    Int13, Reform Jew, Male, 57.
18    Int12_RL, Reform Jew, Male, 50.

of social media. At the individual level, desires to "keep up with the times" are negotiated in tandem with concerns of distractions and temptations on social media that respondents believe may test one's ability to stay on a righteous, moral path. These opportunities and challenges also arise at the community level, such that religious institutions actively seek ways to bridge the perceived generational divide and integrate digital technologies to become more accessible to an even wider community.

Social media technologies have demonstrated the resilience of community. The virtual moral community has the capacity to bring together individuals from broader social circles, who all share similar understandings of faith, sacredness, and meaning. The human affinity for socialization and solidarity still exists without physical space. In many ways, the COVID-19 pandemic illustrates how the mediatization of religious communities is a necessary process to exist in an era of immense social and cultural change. Our respondents' congregations had already begun the shift towards establishing religious community online prior to the pandemic. Social media not only gives religious communities access to a global audience, but, in many ways, it validates religious communities' presence in an increasingly widespread, diverse, and digital society.

This inclusivity, however, comes at the potential expense of traditional boundaries and identity of the religious community. Digital technologies and social media have afforded a new means to expand in-group ties, but not necessarily strengthen them. While there are new ways to reach community members over new mediums (such as live-streamed services or online small groups), those mediums carry the risk of diluting a sense of strong community. Our respondents engage in religion online (Frost and Youngblood 2014; Helland 2000). Even with—and perhaps because of—expanded virtual community, religious people still emphasize the need for embodied presence. Our respondents use digital media as an extension of their physical communities. Even for some subgroups, such as the elderly, who rely on online functions as a substitute for being in physical congregations, there is still a desire to think of online functionality within the context of the physical religious community. Additionally, larger communities have the potential to alienate local community members, as religious leaders struggle to meet the online demands of a broader group. Individual niche needs may become overlooked or not as readily addressed. Yet, the greater access provided by digital technologies such as social media affords religious communities the possibility of increasing participation in local communities while also creating a community for some in the absence of a physical space.

Finally, although many Jews and Muslims identify moral concerns regarding social media, these concerns varied in the degree to which they were directly linked to their faith. For instance, while some expressed concerns about immoral content that may expose children to indecency or immodesty, others expressed more general concerns about the role social media plays in spreading false information or in facilitating cyberbullying. American opinions more broadly are consonant with religious minorities specifically, in showing that social media platforms do not adequately address bullying (Vogels 2021). These patterns demonstrate how, for our respondents, moral tensions were not always centered explicitly within the context of religion. Jewish and Muslim respondents may not always view the challenges of social media in terms of their faith but may situate their moral concerns in the context of broader social ethics.

It is important, however, to note that there were also moral concerns that were unique to each community. While Orthodox Jewish respondents discussed concerns surrounding the way social media resulted in an aggrandizement of oneself that went against notions of modesty, Muslim respondents noted the ways social media notifications disrupted the ritual of prayer. Despite the myriad moral concerns expressed, embracing social media is often posited not only as an inevitability, but a necessity—both to sustain intergenerational connections within the synagogue or mosque and to remain a relevant feature of the evolving socio-cultural US landscape, more broadly.

Our study provides a much-needed empirical addition to the sociological scholarship on religion and social media technologies. Our findings build on existing scholarship

on religion and social media by demonstrating the way that religious people use social media and illustrating how social media technology can change the relationship religious individuals have with ritual aspects of religious communities. Social media may remove some barriers of access between religious leaders and congregants. This has implications for current research that examines the connections between religious practice online and in person (Campbell and Evolvi 2020; Campbell and Vitullo 2016). Our research also builds on literature regarding moral attitudes online. While religious minorities do share specific moral concerns based on their beliefs, most moral concerns are not based solely within religion but often with regards to family and children. This may point to the importance of more research that examines the connections between family, religion, and social media.

Future work would benefit from greater analysis regarding the ways in which spatial context may also shape perspectives on religion and social media (e.g., urban center vs. rural area). Social media may lend well to some contexts over others, depending on a variety of factors including digital infrastructure, population density, and, for religious minorities, ethnic and religious diversity. This study demonstrates additionally that there are unique differences across generations that may have an impact on religious perceptions of social media. While social media may assist the elderly, who may be unable to physically access religious spaces, they may be unable to utilize certain social media platforms because of a lack of digital literacy. Data for this study were collected from 2011 to 2014. Social media has evolved rapidly since then. With the global COVID-19 pandemic, many of the communities we studied were compelled to increase and diversify digital and social media use. Future research should examine new social media technology and the change in use and functionality for religious communities during the global pandemic.

Likewise, while this study focuses on comparing the perspectives of religious minorities across two distinct traditions (Judaism and Islam), we acknowledge that additional variation may arise across denominations. While some differences between Orthodox and Reform Jews emerged in this analysis, more work investigating Muslim Americans and other religious minorities would benefit from unearthing potential heterogeneity within these groups.

Nonetheless, this study contributes to the burgeoning, yet still limited, literature on religion and social media by illuminating some of the ways religious minorities—Jews and Muslims specifically—understand and use social media. While extant literature on the topic tends to overlook minority religious traditions, this research brings to light both the distinct and similar ways that religious minorities view social media as a tool to practice religion, and their moral attitudes toward the platforms as well.

## 4. Materials and Methods

In order to address these research questions, we analyzed 52 in-depth interviews from a previous study focusing on how religious Americans perceive science and science-related technologies. This project involved mixed-methods data collected between 2011 and 2014.

Study participants for the interview study were sampled through several non-random sampling strategies, including key-informant referrals, participant observations, and snowball sampling. As part of the full study, a total of 319 interviews were conducted with Evangelical Protestants, Mainline Protestants, Black Protestants, Jews, and Muslims. The larger data collection included: 40 interviews from three Reform Jewish synagogues, 24 interviews from two Orthodox Jewish synagogues, and 28 interviews from three Sunni Muslim mosques. These respondents included a range of perspectives, including Orthodox Jews, Reform Jews, and Sunni Muslims. Interviews were conducted in Houston and Chicago between 2011 and 2014. For the scope of this project, our analysis centered on the responses of 52 religious minorities (29 Jews and 23 Muslims), comprising both leaders and congregants, and sampled from religious organizations in Houston and Chicago. Respondents were specifically asked questions about the connection between religion and social media.

Most Muslim communities in the United States are Sunni, and many of these communities are Arab and South Asian (Schmidt 2004). Our sample reflects this, with two Arab and 26 South Asian respondents. We understand that the category "Jewish" can refer to both a religious and ethnic category; therefore, for the purposes of this study, we specifically sought out the responses of practicing religious Jews. Interview participants were either recruited by congregational leaders or directly by researchers involved in the study. Other participants were recruited through snowball sampling. Researchers strove to have diversity with regards to gender, age, and socioeconomic status.

Interviews were semi-structured and centered on questions related to the relationship between religion and science. For this article, we focused on the responses from questions related to perceptions of religion and social media. Out of the total number of Jewish and Muslim respondents, 52 respondents were asked the following: "Another topic we are interested in is social media. How do you see social media technologies in relation to your faith? For example, what kind of role do these technologies have on the sense of community within your congregation?" The interviews, on average, lasted for 66 min. Interviews were recorded, transcribed, and then coded for relevant themes using the software ATLAS.ti. After coding for overall themes, we looked for overlap and distinctions in the way Jews and Muslims talked about social media in relation to their specific community dynamics. Although themes mostly overlapped for both communities, we identified the way that certain themes were interpreted differently by some given the faith context.

**Author Contributions:** E.H.E. acquired funding for the study, conceptualized the research study and developed the methodology for the study, contributed to analysis writing, review and editing. J.F. analyzed data and contributed to writing. C.R. contributed to writing of the paper. All authors have read and agreed to the published version of the manuscript.

**Funding:** Data collection for this paper is part of the "Religious Understandings of Science" study, funded by the John Templeton Foundation, Grant #38817, Elaine Howard Ecklund, PI.

**Acknowledgments:** Special thanks to Sharan Kaur Mehta for her time and assistance reviewing this piece.

**Conflicts of Interest:** The authors declare no conflict of interest.

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
