# Peer review of "Navigating Religion Online: Jewish and Muslim Responses to Social Media"

_religions, doi:10.3390/rel12040258_

Round 1

Reviewer 1 Report

Title: Navigating the “Double-Edged Sword:” Jewish and Muslim Responses
to Social Media

The topic of the manuscript is interesting, it fits into the profile of the journal Religions. However, the scientific value of the manuscript is made problematic by the fact that the data collection (52 in-depth interviews) that formed the basis of the research took place long ago, between 2011-2014. Computer mediated communication, social media has undergone a significant transformation since then, and it is likely that the opinions of former respondents may be more changed.

Helland’s distinction between religion online and online religion is not reflected in the study,  but this would be important for research.

The manuscript states that less research has examined on the topic (moral attitudes around social media for Jews and Muslims and transforming the sense of religious community), but a large number of relevant research materials have emerged in recent years that need to be incorporated into the study (regarding religious communities: Campbell 2013, 2018, 2019 Cheong 2012, the study of Jewish communities was conducted by Oren Golan (Golan 2018, 2019). These recent literatures provide an indispensable scientific background for such a study.

The text contains mistyping in some places (Hjaryard), there are formal errors in the references (Dawn N. Norris, Petter Törnberg).

Author Response

Dear Editorial Team,

We appreciate your invitation to revise and resubmit our manuscript, “Navigating the Double-Edged Sword:” Jewish and Muslim Responses to Social Media” (Manuscript ID: religions-1126007). We are grateful for the reviewers’ affirmation of our work and helpful suggestions for further revision. Below we list the reviews in plain type and how we responded to each section of them in italics. Please note that all specific line references correspond to line numbers when viewing in “all mark up” mode, which shows tracked changes.

The topic of the manuscript is interesting, it fits into the profile of the journal Religions. However, the scientific value of the manuscript is made problematic by the fact that the data collection (52 in-depth interviews) that formed the basis of the research took place long ago, between 2011-2014. Computer mediated communication, social media has undergone a significant transformation since then, and it is likely that the opinions of former respondents may be more changed.

We appreciate the reviewers comment regarding the time of data collection and how social media has rapidly evolved since then. We are unable to add more interviews to our study at this time. We do, however, add a note about these limitations in the discussion section on lines 535-540.

Helland’s distinction between religion online and online religion is not reflected in the study, but this would be important for research.

We include this and other relevant references on lines 85-97 and 482-488.

The manuscript states that less research has examined on the topic (moral attitudes around social media for Jews and Muslims and transforming the sense of religious community), but a large number of relevant research materials have emerged in recent years that need to be incorporated into the study (regarding religious communities: Campbell 2013, 2018, 2019 Cheong 2012, the study of Jewish communities was conducted by Oren Golan (Golan 2018, 2019). These recent literatures provide an indispensable scientific background for such a study.

We appreciate these recommended articles and have included them and other references on lines 30-31, 63-64, 104-105, and 520-522.

The text contains mistyping in some places (Hjaryard), there are formal errors in the references (Dawn N. Norris, Petter Törnberg)."

We have corrected these errors on lines 31, 179, and 655.

Reviewer 2 Report

This paper has potential to add meaningful insights to the study of religion and media (social media). The interview excerpts are interesting and raise some useful questions about the digital divide, community, and social norms within Judaism and Muslim minorities in the U.S.

There are several areas that need to be strengthened. First, the study needs to address more recent scholarship on 'digital religion.' (Heidi Campell's most recent work, Tim Hutchings' work, the work of Ruth Tsuria, and others, for example). This will help the scholars get to a more nuanced approach. Right now, the study does not add enough unique insights to push existing research forward. Scholars have long said that religions have an ambivalent relationship with technology - this double-edged sword dynamic is very old, and simply reiterating it is not itself enough of a contribution to justify publishing the piece as it is. The authors need to engage more specific questions, such as -- what is it about being a religious minority in the United States that might shape how these respondents engage with social media? What is unique to their faith traditions that impacts their relationship with social media? Why is it useful to focus on minority Muslims and Jews beyond the fact that they are minorities? What is it about their location in the U.S. that might impact their perspective? These and other similar questions need to be carefully considered. Right now, the study feels quite underdeveloped.

A few additional minor suggestions/thoughts: 1) I'm not convinced that using religion-science to help frame this study is a useful strategy. It seems more of a distraction than anything. The study is clearly about religion morals and social media; introducing bigger theoretical questions about religion-science do not seem to fit, nor are they done without enough clarity for them to be justified. 2) These interviews are between 7 and 10 years ago, but nowhere do the authors mention how social media have changed since this time (a lot has changed!). 3) I'd like to see more interview excerpts to get a richer qualitative sense of interviewees' responses.

Author Response

Dear Editorial Team,

We appreciate your invitation to revise and resubmit our manuscript, “Navigating the Double-Edged Sword:” Jewish and Muslim Responses to Social Media” (Manuscript ID: religions-1126007). We are grateful for the reviewers’ affirmation of our work and helpful suggestions for further revision. Below we list the reviews in plain type and how we responded to each section of them in italics. Please note that all specific line references correspond to line numbers when viewing in “all mark up” mode, which shows tracked changes.

There are several areas that need to be strengthened. First, the study needs to address more recent scholarship on 'digital religion.' (Heidi Campell's most recent work, Tim Hutchings' work, the work of Ruth Tsuria, and others, for example). This will help the scholars get to a more nuanced approach. Right now, the study does not add enough unique insights to push existing research forward. Scholars have long said that religions have an ambivalent relationship with technology - this double-edged sword dynamic is very old, and simply reiterating it is not itself enough of a contribution to justify publishing the piece as it is. The authors need to engage more specific questions, such as -- what is it about being a religious minority in the United States that might shape how these respondents engage with social media? What is unique to their faith traditions that impacts their relationship with social media? Why is it useful to focus on minority Muslims and Jews beyond the fact that they are minorities? What is it about their location in the U.S. that might impact their perspective? These and other similar questions need to be carefully considered. Right now, the study feels quite underdeveloped.

We thank the reviewers for their suggestions in literature. We have added these references and others in lines 90-95, 102-105, and 135-138. We have also taken your feedback about the “double-edged sword” seriously and have decided to remove it from our paper title. We instead focus the complexity that social media has in both encouraging religious community for our respondents and compelling respondents to define their moral boundaries and the ways that those who are Jewish and Muslim do this, in particular, which we argue is necessary to examine given the more general Christian community focus of the existing literature. We include more literature regarding how social media impacts religious minorities specifically on lines 42-44, 109-118, and 132-138. We also add on lines 16-18 an imperative for why we focus on Jews and Muslims.

A few additional minor suggestions/thoughts: 1) I'm not convinced that using religion-science to help frame this study is a useful strategy. It seems more of a distraction than anything. The study is clearly about religion morals and social media; introducing bigger theoretical questions about religion-science do not seem to fit, nor are they done without enough clarity for them to be justified.

In response we have framed the paper a bit more to reflect the impact of social media on community, and the impact of religion on moral attitudes towards social media. While our interviews come from data collected on American perceptions of science-related technologies, the analysis focuses on how the Jewish and Muslim individuals and leaders we interviewed understand social media in relation to morality and community. We hope that this is demonstrated in the additions we have added to the literature review, discussion, and methods section.

2) These interviews are between 7 and 10 years ago, but nowhere do the authors mention how social media have changed since this time (a lot has changed!).

We have updated the Discussion section on lines 535-540 to mention how social media has rapidly evolved since the time of our interviews.

3) I'd like to see more interview excerpts to get a richer qualitative sense of interviewees' responses.

While we appreciate the reviewers’ suggestion, at this time we are unable to add additional interview material.

Reviewer 3 Report

The article could be interesting if it has more solid references and the sample is expanded. At the current stage, 52 surveys in 2 cities in the US are still not enough to draw conclusions.

Very broad article: who are "Jews"? from which communities? And "Muslims" as well. Readers need to understand whom you as authors are analyzing. Too general sentences make this unclear: have you talked to leaders? To a simple believer?

References to the Pew Religion are old (half of US 33 adults see religion shared regularly by others online (Pew Research Center 2014). Authors could check more recent reports from the same organization that could help them to better frame the article.

In page 4 this sentence needs more explanation: is not clearly undesrstood academically speaking what the author/s understand by "temptations: acknowledging potential moral distractions and temptations. 

References that could enrich the paper:

Apart from Heidi Campbells new published articles and books, as well as Helland's ones.

https://www.tandfonline.com/doi/full/10.1080/23753234.2017.1347800

https://www.tandfonline.com/doi/abs/10.1080/0048721X.2020.1754605

https://heiup.uni-heidelberg.de/journals/index.php/religions/article/view/23945

https://www.tandfonline.com/doi/abs/10.1080/14755610.2011.579714

https://www.tandfonline.com/toc/hjmr20/current

https://journals.sagepub.com/doi/10.1177/1461444812462848

Author Response

Dear Editorial Team,

We appreciate your invitation to revise and resubmit our manuscript, “Navigating the Double-Edged Sword:” Jewish and Muslim Responses to Social Media” (Manuscript ID: religions-1126007). We are grateful for the reviewers’ affirmation of our work and helpful suggestions for further revision. Below we list the reviews in plain type and how we responded to each section of them in italics. Please note that all specific line references correspond to line numbers when viewing in “all mark up” mode, which shows tracked changes.

The article could be interesting if it has more solid references and the sample is expanded. At the current stage, 52 surveys in 2 cities in the US are still not enough to draw conclusions.

In response we have significantly expanded the literature cited and wrestle with that literature. The study is not meant to be a quantitative study – the reviewer’s use of the word “surveys” leaders us to believe they may see the study as making quantitative claims; we do argue in the paper that understanding how Jewish and Muslim congregants and congregational leaders narrate their relationship to social media is an important contribution to the existing literature.

Very broad article: who are "Jews"? from which communities? And "Muslims" as well. Readers need to understand whom you as authors are analyzing. Too general sentences make this unclear: have you talked to leaders? To a simple believer?

We include a note in the methods section lines 558-570 regarding our sample. We interview both religious leaders and congregants. In lines 556-558 we note that our respondents include Orthodox and Reform Jews and Sunni Muslims.

References to the Pew Religion are old (half of US 33 adults see religion shared regularly by others online (Pew Research Center 2014). Authors could check more recent reports from the same organization that could help them to better frame the article.

We have updated Pew Research Center references on lines 29, 35, 179, and 501.

In page 4 this sentence needs more explanation: is not clearly understood academically speaking what the author/s understand by "temptations: acknowledging potential moral distractions and temptations.

Thank you for this comment. We have revised this sentence on lines 214-218.

References that could enrich the paper:

Apart from Heidi Campbells new published articles and books, as well as Helland's ones.

https://www.tandfonline.com/doi/full/10.1080/23753234.2017.1347800

https://www.tandfonline.com/doi/abs/10.1080/0048721X.2020.1754605

https://heiup.uni-heidelberg.de/journals/index.php/religions/article/view/23945

https://www.tandfonline.com/doi/abs/10.1080/14755610.2011.579714

https://www.tandfonline.com/toc/hjmr20/current

https://journals.sagepub.com/doi/10.1177/1461444812462848

We have updated the literature review to include these references.

Round 2

Reviewer 1 Report

I propose to accept the study in its present form.

Reviewer 2 Report

Thank you for making changes to the manuscript. 

Reviewer 3 Report

Suggestions have been added and make the article an interesting consistent paper.